# Establishment of Replication Deficient Vesicular Stomatitis Virus for Studies of PEDV Spike-Mediated Cell Entry and Its Inhibition

**DOI:** 10.3390/microorganisms11082075

**Published:** 2023-08-12

**Authors:** Huaye Luo, Lilei Lv, Jingxuan Yi, Yanjun Zhou, Changlong Liu

**Affiliations:** 1Shanghai Veterinary Research Institute, Chinese Academy of Agricultural Sciences, Shanghai 200241, China; luohuaye19971115@163.com (H.L.); 18251892518@163.com (L.L.); yijingxuan7@163.com (J.Y.); yjzhou@shvri.ac.cn (Y.Z.); 2Jiangsu Co-Innovation Center for the Prevention and Control of Important Animal Infectious Disease and Zoonosis, Yangzhou University, Yangzhou 225009, China

**Keywords:** porcine epidemic diarrhea virus, pseudovirus, VSV, virus entry, virus receptor

## Abstract

The porcine epidemic diarrhea virus (PEDV) is a highly contagious and virulent enteric coronavirus that causes severe enteric disease in pigs worldwide. PEDV infection causes profound diarrhea, vomiting, and dehydration in pigs of all ages, resulting in high mortality rates, particularly among neonatal piglets. The spike glycoprotein (S) of PEDV plays a crucial role in binding to the host cell receptor and facilitating fusion between the viral and host membranes. Pseudotyped viral particles featuring the PEDV S protein are valuable tools for investigating virus entry, identifying neutralizing antibodies, and developing small molecules to impede virus replication. In this study, we used a codon-optimized PEDV S protein to generate recombinant pseudotyped vesicular stomatitis virus (VSV) particles (rVSV-ΔG-EGFP-S). The full-length S protein was efficiently incorporated into VSV particles. The S protein pseudotyped VSV exhibited infectivity towards permissive cell lines of PEDV. Moreover, we identified a new permissive cell line, JHH7, which showed robust support for PEDV replication. In contrast to the SARS-CoV-2 spike protein, the removal of amino acids from the cytoplasmic tail resulted in reduced efficiency of viral pseudotyping. Furthermore, we demonstrated that 25-hydroxycholesterol inhibited rVSV-ΔG-EGFP-S entry, while human APN facilitated rVSV-ΔG-EGFP-S entry through the use of ANPEP knockout Huh7 cells. Finally, by transducing swine intestinal organoids with the rVSV-ΔG-EGFP-S virus, we observed efficient infection of the swine intestinal organoids by the PEDV spike-pseudotyped VSV. Our work offers valuable tools for studying the cellular entry of PEDV and developing interventions to curb its transmission.

## 1. Introduction

Porcine epidemic diarrhea virus (PEDV) is a highly pathogenic virus that causes severe enteric disease in swine globally [1]. PEDV infection results in severe diarrhea, vomiting, and dehydration in pigs of all ages, with high mortality rates observed in neonatal piglets [2]. Since 2010, outbreaks caused by novel, highly pathogenic strains of PEDV have resulted in significant economic losses in the global pig industry [3]. PEDV is an alphacoronavirus with a single-stranded, positive-sense RNA genome. The genome spans approximately 28 kbp and harbors at least seven open reading frames (ORFs) that encode various viral proteins. The ORF1a and ORF1b regions are responsible for encoding the viral replicase, while the remaining ORFs encode structural proteins, including the spike (S), envelope (E), membrane (M), and nucleocapsid (N) proteins [4,5]. The S protein of PEDV is responsible for binding to specific receptors on the surface of targeting cells, enabling viral entry and subsequent infection. It is a type I membrane glycoprotein comprising an N-terminal signal peptide, a large extracellular region, a single transmembrane domain, and a short cytoplasmic tail. Similar to other coronavirus S proteins, the ectodomain of PEDV S can be divided into two domains: the N-terminal S1 subunit, responsible for receptor binding, and the C-terminal S2 subunit, responsible for membrane fusion [6].

Receptor interaction plays a crucial role in determining the cell and tissue tropism of coronaviruses, as well as their pathogenesis and ability to cross species barriers. The spike proteins of coronaviruses possess the capability to bind to various proteinaceous and carbohydrate cell surface molecules, such as carcinoembryonic antigen-related cell adhesion molecule 1 (CEACAM1), angiotensin-converting enzyme 2 (ACE2), aminopeptidase N (APN), and dipeptidyl peptidase 4 (DPP4) [7,8,9,10,11]. The PEDV is known to utilize the porcine aminopeptidase N protein, also known as CD13, as its receptor [12,13]. However, the utilization of pAPN as a receptor for PEDV has been subject to doubt [14,15,16,17]. Although several proteins, including sialic acid, heparan sulfate, occludin, and transferrin [12,18,19,20,21], have been reported to potentially facilitate PEDV entry, the precise mechanism by which PEDV enters cells remains unclear. Hence, further research is warranted to gain a comprehensive understanding of the PEDV entry process.

The binding affinity of PEDV to cellular receptors displays a promiscuous nature. PEDV has the capacity to infect various cell lines derived from different species, including bats and primates [12,22]. Comparative analysis of the genomic sequence of PEDV reveals a closer relationship to bat alpha coronaviruses than to other viruses within the same genus, suggesting that interspecies transmission between bats and pigs may have occurred through an intermediate host. The ability of PEDV to infect cells across different species implies that the virus, similar to MERS-CoV [9], utilizes evolutionarily conserved host cell components as receptors, thereby increasing the potential for cross-species and potentially zoonotic transmission.

Vesicular stomatitis virus (VSV) belongs to the Rhabdoviridae family and possesses a characteristic bullet shape along with a negative-strand RNA genome. This virus encompasses five structural genes that encode distinct proteins: nucleocapsid (N), phosphoprotein (P), matrix protein (M), glycoprotein (G), and large protein (L) [23]. The genome of VSV is relatively simple, rendering it amenable to manipulation for scientific research purposes [24]. One notable feature of VSV is its capacity to incorporate and tolerate foreign envelope proteins on its surface, making it an excellent model for investigating the function of glycoproteins from other viruses [25]. 

In this study, we generated a replication-deficient virus, rVSV-ΔG-EGFP-S, expressing the EGFP marker by incorporating the PEDV S protein into VSV. Through our experiments, we observed that the recombinant virus displayed a remarkable ability to successfully infect PEDV-susceptible cell lines and organoids. Additionally, we identified a new cell line that supports PEDV replication using PEDV spike-pseudotyped virus. These findings highlight the potential of the pseudotyped virus as a valuable tool for investigating the entry mechanism of PEDV.

## 2. Materials and Methods

### 2.1. Ethics Statement

The recombinant replication-deficient VSV studies were conducted under biosafety level 2 (BSL2) conditions.

### 2.2. Plasmids

A human codon-optimized full-length spike (S) gene for the SD isolate of PEDV (GenBank accession no. MZ596343) was synthesized by Saiheng Biotech (Shanghai, China) and inserted into the pCAGGS vector using EcoRⅠ and XhoⅠ restriction enzyme sites to get pCAGGS-PEDV-S. The fragments of the cytoplasmic tail truncation of the spike gene were generated by PCR using primers in Table 1 and then inserted into the pCAGGS vector. A VSV-G protein expression plasmid, pMD2.G, was obtained from Addgene (cat#:12259, Watertown, MA, USA). A cDNA clone of VSV (pVSV-ΔG-EGFP) containing expression cassettes for enhanced GFP (EGFP) in place of the VSV-G gene was purchased from Vector Builder (Guangzhou, China). Helper plasmids pBS-N, pBS-P, pBS-L, and pBS-G were purchased from Kerafast (Boston, MA, USA).

### 2.3. Cell Lines, Viruses and Antibody

BSR-T7 (kindly provided by Prof. Xusheng Qiu), Vero (ATCC, CCL-81), Huh7 (kindly provided by Dr. Rong Zhang at Fudan University), JHH7 (Procell, Wuhan, China), HEK-293T (ATCC, CRL-3216), HeLa (ATCC, CCL-2), BHK-21 (Procell, Wuhan, China), Li7 (Procell, Wuhan, China), and MDCK (kindly provided by Prof. Yingjie Sun) were cultured in DMEM (Gibco, Shanghai, China) containing 10% fetal bovine serum (FBS) and 1% penicillin/streptomycin at 37 °C with 5% CO_2_. The Huh7, JHH7, and Li7 cell lines, derived from human hepatocellular carcinoma, exhibit an epithelial-like morphology. The BHK-21 cell line originates from a baby hamster’s kidney. The MDCK (Madin-Darby Canine Kidney) cell line is widely used in cell biology and virology research and is derived from the kidney of a female cocker spaniel. BSR-T7 is a continuous cell line derived from African green monkey kidney cells (BS-C-1), which stably expresses T7 RNA polymerase. Vero cells are a continuous line of cells derived from the kidney of an African green monkey (Cercopithecus aethiops). HEK-293T cells are a cell line derived from human embryonic kidney (HEK) cells by introducing the SV40 large T antigen gene. Recombinant vaccinia virus vTF-7.3 was kindly provided by Prof. Weike Li, Lanzhou Veterinary Research Institute, CAAS. The virus stock of the PEDV SD strain (GenBank accession No.MZ596343) was prepared and titrated on the Vero cells by TCID_50_. A monoclonal antibody against PEDV S protein was purchased from ZhaoRui Biotech (cat#: M100047, Shanghai, China). A monoclonal antibody for human ANPEP (cat#: ab108310) was obtained from Abcam. HRP-linked secondary antibodies for mouse IgG (cat#: 7076) and rabbit IgG (cat#: 7074) were purchased from Cell Signaling Technology (Danvers, MA, USA).

### 2.4. Recovery and Production of rVSV-ΔG-EGFP-G

Recombinant rVSV-ΔG-EGFP-G was rescued using previously established methods [24,26]. In brief, BSR-T7 cells were infected with vTF-7.3 at an MOI of 5 in a T25 flask. After 1 h, the cells were co-transfected with the pVSV-ΔG-EGFP plasmids (4 μg) together with the four helper plasmids pBS-N (4 μg), pBS-P (4 μg), pBS-L (2 μg), and pBS-G (3 μg), which encode VSV N, P, L, and G proteins, using Lipofectamine 3000. Forty-eight h post-transfection, the supernatant containing the rVSV-ΔG-EGFP-G was collected, centrifuged at 2000× *g* for 10 min to remove cell debris, and filtered through a 0.22 μm filter. For rVSV-ΔG-EGFP-G production, HEK-293T cells were seeded in a T25 flask. When reaching 80% confluence, the cells were transfected with 7 μg of the pMD2.G plasmid using the calcium phosphate method [27,28]. At 24 h post-transfection, the cells were infected with 2 mL of filtered supernatant from the primary transfection. At 48 h post-infection, the supernatant was collected, centrifuged, and used for sequential passage in HEK-293T cells that were transfected with the pMD2.G plasmid. 

### 2.5. Production of PEDV S Pseudotyped rVSV-ΔG-EGFP-S and Infection

To produce PEDV S pseudotyped VSV, approximately 3 × 10^6^ HEK-293T cells were seeded in a T75 flask and transfected with 24 μg of pCAGGS-PEDV-S or C-terminal truncated plasmids 24 h later using lipofectamine 3000. The cells were incubated at 37 °C with 5% CO2 for 6 h. Then the cells were infected with 15 mL of DMEM containing 1 × 10^6^ TCID_50_ of rVSV-ΔG-EGFP-G. Following 1 h incubation, the virus was removed, and the cells were washed five times with DPBS. Subsequently, 15 mL of DMEM containing 2% FBS and 1% penicillin/streptomycin was added to the flask. The supernatants were collected 48 h post-infection, centrifuged at 2000× *g* for 10 min to remove cell debris, filtered through a 0.22 μm filter, and stored at −80 °C.

### 2.6. 25-Hydroxycholesterol (25HC) Treatment and PEDV Pseudovirus Infection

Huh7 cells were plated in 6-well plates, pre-treated with 25HC for 2 h, and then infected with PEDV pseudovirus at a MOI of 0.05 in DMEM without 25HC. After 1 h of incubation, the virus was removed and fresh medium (DMEM with 2% FBS) was added. The cells were then cultured at 37 °C for 12 h. EGFP-positive cells were evaluated by capturing pictures with an Eclipse Ts2R-FL microscope (Nicon, Japan) and infection rates were determined by analyzing GFP levels using a CytoFLX flow cytometer (Beckman Coulter, Brea, CA, USA).

### 2.7. Flow Cytometry Analysis

Flow cytometry was employed to determine the percentage of infected cells with pseudotyped viruses. In brief, virus-infected cells were rinsed with DPBS, treated with Trypsin/EDTA, and suspended in 1 mL of fluorescence-activated cell sorting (FACS) buffer (PBS containing 2 mM EDTA, 0.1% sodium azide, and 2% FBS). GFP-positive cells were determined using a CytoFLX flow cytometer (Beckman Coulter, Brea, CA, USA). The acquisition was set for 60,000 events per sample. The data were analyzed using flowJo software (V10, Ashland, OR, USA). All samples were prepared in triplicate.

### 2.8. ANPEP Knockout in Huh7

ANPEP-KO Huh7 cells were generated as previously described [27,28,29]. Briefly, Huh7 cells were transfected with a pair of small guide RNAs (Table 1) in the LentiCRISPRv2-puro (cat#:98290, Addgene, Watertown, MA, USA) vector using Lipofectamine 3000 (Life Technologies, Carlsbad, CA, USA). After 24 h, the fresh medium with 3 μg/mL puromycin was added and incubated for 3 days. Single-positive cells were then manually diluted in 96-well plates. Single-cell clones were expanded and validated by western blotting and sequencing using the primers shown in Table 1.

### 2.9. RT-qPCR for PEDV Titer

RNA was purified from the supernatant containing PEDV with the TRIzol reagent. Subsequently, cDNA was synthesized using SuperScriptTM III (Invitrogen, Shanghai, China) according to the manufacturer’s instructions. qPCR was performed using SYBR^®^ Advantage^®^ qPCR Premix (Clontech, Mountain View, CA, USA. cat#: 639676) and a ViiA™ 7 Real-Time PCR System instrument (Applied Biosciences, Waltham, MA, USA) The primer sequences for the PEDV N gene are provided in Table 1.

### 2.10. Western blotting

Cell lysates were prepared using RIPA buffer supplemented with a protease inhibitor cocktail (cat#: P8340, Sigma, St. Louis, MI, USA). The cell lysates or viruses in supernatants mixed with 5 × loading buffer were boiled at 100 °C for 10 min and separated by electrophoresis in a 7.5% or 10% SDS-PAGE gel. The blot was blocked with TBST (10 mM Tris-HCl, pH 7.5, 150 mM NaCl, and 0.1% Tween-20) containing 5% non-fat dry milk and then incubated with the primary antibody in TBST at 4 °C overnight. After washing with TBST, the membrane was incubated with a horseradish peroxidase (HRP)-conjugated secondary antibody for 1 h at room temperature with constant agitation. Signals were raised with a hypersensitive ECL chemiluminescence kit (cat#:P10100, NCM Biotech, Suzhou, China) and detected using a ChemiDoc™ MP imaging system (Bio-Rad, Hercules, CA, USA). 

### 2.11. Porcine Intestinal Organoid Culture and Transduction

Porcine intestinal organoids from 10-day pig intestine were cultured as previously described [29]. Briefly, the frozen porcine intestinal organoids were recovered from liquid nitrogen. Then, the organoids were resuspended with matrigel and plated in droplets of 50 µL per well of a 24-well plate. After allowing the matrigel to solidify, organoids culture medium (Advanced DMEM/F12, 100 µg/mL primocin, 2 mM GlutaMax, 10 mM HEPES, 1 × N_2_, 1 × B27, 1 mM N-Acetyl-Cysteine, 10 mM Nicotinamide, 50% L-WRN conditioned medium, 50 ng/mL human EGF, 500 nM A83-01, and 3 μM SB202190) supplemented with 10 μM ROCK inhibitor was added to the plates, and organoids were cultured at 37 °C, 5% CO_2_.

Organoid transduction was conducted according to the method previously described [29]. Briefly, the organoids were collected in cold Advanced DMEM/F12 (AD) medium and then washed once before being digested using TrypLE (Invitrogen, Shanghai, China). After digestion, the organoids were washed once again in AD medium. The rVSV-ΔG-EGFP-S infection was performed at a MOI of 0.1 in AD medium. Following incubation for 2 h at 37 °C 5% CO_2_, the cultures were washed twice with excess AD medium to eliminate unbound virus. The organoids were re-embedded into 50 μL of matrigel in a 24-well plate and cultured in 500 μL of culture medium.

### 2.12. Statistical Analyses

The data analysis was conducted using GraphPad Prism 9 (GraphPad, San Diego, CA, USA). The data are expressed as the mean ± standard deviation (SD) of at least three replicates. The unpaired Student’s *t*-test was performed to calculate the *p*-values. The significance level (*p*-value) was set at <0.05 (*), <0.01 (**), and <0.001 (***).

## 3. Result

### 3.1. Generation of Replication-Deficient PEDV S Pseudotyped rVSV-ΔG-EGFP-S

The backbone of the pseudotyped virus used in this study was based on the VSV virus, with the G gene replaced with the EGFP reporter gene. To generate the PEDV S pseudotyped VSV, a VSV pseudotyped virus packaging system [24,25] was employed, which has previously been utilized for the production of various pseudotyped viruses, including SARS-CoV [30], SARS-CoV-2 [31,32], Ebola virus [33], Oropouche virus [34], Nipah virus [35], and Rift Valley fever virus [36]. The schematic procedure for generating the PEDV S pseudotyped virus is depicted in Figure 1A. HEK-293T cells were transfected with the PEDV S protein expression plasmid and subsequently infected with the VSV G pseudotyped virus (rVSV-ΔG-EGFP-G). The S protein from PEDV was then incorporated as the membrane protein on the surface of the VSV virus (rVSV-ΔG-EGFP-S). To verify the incorporation of the spike protein, the surface protein in the PEDV pseudovirus was detected using Western blotting with a PEDV S mono-antibody. The specific bands were observed in the lanes of the PEDV pseudovirus supernatant, while no specific band was detected in the control (Figure 1B). 

Upon incubation with cells expressing PEDV receptors, the rVSV-ΔG-EGFP-S viruses attach to and enter the cells through the receptors on the cell surface. Since Vero cells were originally employed for the isolation of PEDV, we then infected the Vero cells with the pseudotyped virus rVSV-ΔG-EGFP-S. The fluorescence microscope analysis demonstrated infection of Vero cells by the PEDV S pseudotyped virus rVSV-ΔG-EGFP-S (Figure 1C). Since the pseudotyped virus does not express surface proteins, it cannot form virus particles containing surface proteins. Consequently, the pseudotyped rVSV-ΔG-EGFP-S virus is unable to replicate sequentially and only exhibits single-round infection capability.

### 3.2. Characterizing the Entry of rVSV-ΔG-EGFP-S on Several Cell Lines

After successfully generating the PEDV S pseudotyped virus rVSV-ΔG-EGFP-S, our next objective was to assess its entry into various cell lines. Previous reports have indicated that PEDV is capable of infecting multiple cell lines derived from different species, such as Vero and Huh7 [22,37]. We included six cell lines in our investigation: Huh7, BHK-21, MDCK, HeLa, JHH7, and Li7. Each cell line was exposed to an equal amount of rVSV-ΔG-EGFP-S virus (0.05 MOI). The results revealed a significant susceptibility of Huh7 cells to the rVSV-ΔG-EGFP-S virus, consistent with prior findings of PEDV infection [12,22]. As expected, BHK-21, HeLa, MDCK, and Li7 cells displayed resistance to the rVSV-ΔG-EGFP-S virus (Figure 2A). Notably, JHH7 cells exhibited a high susceptibility to rVSV-ΔG-EGFP-S and produced robust signals (Figure 2A). To confirm the permissiveness of JHH7 cells to PEDV, we subsequently infected them with the PEDV SD strain, which demonstrated strong replication support in JHH7 cells (Figure 2B). Collectively, these findings suggest that the PEDV S pseudotyped VSV can serve as a versatile tool for studying PEDV entry in vitro.

### 3.3. C-Terminal Truncation of PEDV Spike Decreases the Efficiency of Viral Pseudotyping

Truncating the cytoplasmic tail of the spike protein in SARS-CoV or SARS-CoV-2 has been shown to enhance the production of pseudotyped lentiviral and VSV vectors [38,39,40,41]. This enhancement is thought to be due to improved incorporation of the spike protein. To investigate the impact of cytoplasmic tail truncation of the PEDV S protein on pseudotyped VSV production, we generated several truncations that removed 7, 10, 19, 20, and 30 amino acids from the cytoplasmic tail of the PEDV S protein, respectively (Figure 3A). The truncated PEDSV S proteins were used to generate pseudotyped viral particles, which were then used to infect Vero cells, as illustrated in Figure 3B. The percentage of EGFP-positive cells was determined by flow cytometry (Figure 3C). In contrast to SARS-CoV and SARS-CoV-2, our experimental results revealed that truncating the PEDV S protein actually decreased the production of pseudotyped VSV. Among the five truncations tested, the truncations with 7 and 10 amino acids deleted exhibited a slight decrease in virus titer. However, the truncation with 19 and 20 amino acids deleted led to a significant reduction in virus titer. Moreover, it was observed that pseudotyped VSV production was almost completely abolished when the truncation involved 30 amino acids (Figure 3B,C). Collectively, these findings indicate that truncation of the cytoplasmic tail of the PEDV spike protein reduces the efficiency of viral pseudotyping.

### 3.4. 25-Hydroxycholesterol Inhibits the Pseudovirus rVSV-ΔG-EGFP-S Entry

The molecule 25-hydroxycholesterol (25HC) has been recognized for its potential to inhibit the entry and replication of various coronaviruses, including PEDV and SARS-CoV-2. This inhibitory activity is mainly achieved by interfering with viral fusion and entry into host cells and disrupting viral protein maturation [42,43,44,45,46]. To further investigate the effect of 25HC on pseudovirus rVSV-ΔG-EGFP-S infection, we pretreated Huh7 cells with 25HC at concentrations of 50 μM and 100 μM for 2 h and subsequently infected these cells with rVSV-ΔG-EGFP-S virus. The samples were collected at 12 hpi and analyzed for GFP expression by fluorescent microscopy (Figure 4A) and flow cytometry (Figure 4B). The result showed that rVSV-ΔG-EGFP-S infection was significantly reduced in both 50 μM and 100 μM 25HC-treated Huh7 cells (Figure 4C). These findings suggest that rVSV-ΔG-EGFP-S could serve as a valuable tool for studying the inhibition of PEDV entry.

### 3.5. Human ANPEP Facilitates rVSV-ΔG-EGFP-S Entry

Previous studies have indicated that PEDV utilizes both porcine and human aminopeptidase N (APN) as receptors to bind to the host cell membrane, which facilitates the virus’s entry into the cell [12]. To confirm previously reported observations using PEDV S rVSV-ΔG-EGFP-S, we generated ANPEP knockout Huh7 cell lines through CRISPR/Cas9 genome editing technology, as previously described [27,28]. Cas9 protein-encoding vectors and two small guide RNAs (sgRNA) targeting exon 1 of the human ANPEP gene (Figure 5A) were transfected into Huh7 cells. Single clones were collected, and the absence of both alleles of the deleted region was confirmed by PCR using primers flanking the deleted region. The ANPEP gene locus of ANPEP knockout Huh7 cells (ANPEP-KO) was further confirmed by DNA sequencing. The ANPEP protein was visualized in the ANPEP-KO Huh7 cells and the parental Huh7 by Western blotting, confirming the absence of the ANPEP protein in the ANPEP-KO Huh7 cells (Figure 5B). 

To determine whether the limitation of PEDV replication in ANPEP-deficient cells is caused by the inhibition of PEDV entry, we infected ANPEP-WT and ANPEP-KO cells with the rVSV-ΔG-EGFP-S virus. At 12 hpi, samples were collected and assessed for GFP expression using fluorescent microscopy. rVSV-ΔG-EGFP-S infection was significantly reduced in ANPEP KO cells (Figure 5C). Furthermore, the virus titers were evaluated with the PEDV SD strain at different time points by qPCR. The virus titers in ANPEP-KO Huh7 cells were observed to decrease by approximately 10-fold in comparison to ANPEP-WT Huh7 cells at both 24 and 48 h (Figure 5D). These findings indicate that ANPEP plays a significant role in the entry of PEDV into host cells. However, ANPEP may not function as a direct receptor for PEDV entry.

### 3.6. rVSV-ΔG-EGFP-S Infects Swine Intestinal Organoids

Organoids have become a valuable tool for investigating the interactions between pathogens and host cells. These 3D cell cultures mimic the architecture and function of specific organs or tissues, providing a more physiologically relevant environment compared to traditional two-dimensional cell cultures [47]. Our group and others have developed various culture systems for porcine intestinal organoids, which have enabled the study of porcine enteric coronaviruses, including PEDV and TGEV, in vitro [29,48,49]. We subsequently evaluated the susceptibility of swine intestinal organoids to the rVSV-ΔG-EGFP-S pseudovirus. The pseudotyped rVSV-ΔG-EGFP-S virus was used to infect swine intestinal organoids. The result showed that the rVSV-ΔG-EGFP-S virus was able to efficiently infect the cells, as evidenced by GFP expression throughout the organoid within 24 h (Figure 6). This observation further solidifies the utility of PEDV S pseudotyped VSV as an invaluable tool for investigating virus entry.

## 4. Discussion

PEDV is a highly contagious coronavirus that infects pigs, causing severe diarrhea and high mortality rates in piglets. Understanding the cellular entry of PEDV is crucial for developing effective treatments and preventive measures. Cellular entry of PEDV occurs through the binding of its spike protein (S protein) to a host cell receptor [50]. The S protein mediates virus attachment to the target cell and subsequent membrane fusion, enabling the virus to enter the cell [6]. Pseudotype viruses bearing heterologous viral glycoproteins can be powerful tools for investigating the entry of many viral pathogens because of their safety and versatility. Lentivirus-based PEDV pseudoviruses have been employed to investigate virus entry. However, the efficiency of lentiviral pseudotyping is considerably limited [12,37]. In this study, we produced a replication-deficient VSV that efficiently encapsulated the PEDV S protein into viral particles. The rVSV-ΔG-EGFP-S exhibited significant infectivity in susceptible cells, and viral infection could be evaluated by observing GFP-expressing cells.

Previous reports have indicated that PEDV can infect multiple cell lines derived from different species [12,22,37,51]. In this study, the susceptibility of six cell lines, including Huh7, BHK-21, MDCK, HeLa, JHH7, and Li7, to the PEDV S pseudotyped VSV was assessed. Among the tested cell lines, Huh7 and JHH7 cells were susceptible to the PEDV spike pseudotyped virus, whereas BHK-21, HeLa, MDCK, and Li7 cells showed resistance. Interestingly, JHH7 cells exhibited susceptibility to the PEDV S pseudotyped VSV, which was first reported. This finding is significant because it suggests that JHH7 cells may be useful for studying the entry mechanism of PEDV. The permissiveness of JHH7 cells to PEDV was subsequently confirmed with the PEDV SD strain, demonstrating strong replication support. Porcine intestinal organoids are three-dimensional structures that mimic the architecture and function of the pig intestine. Researchers have utilized porcine intestinal organoids to study the interaction between PEDV and the intestinal epithelium [29,48,49]. We successfully infected the swine intestinal organoids with rVSV-ΔG-EGFP-S. All these results highlight the versatility of the PEDV S pseudotyped VSV as a tool for investigating PEDV entry dynamics across different cell lines.

The first identified cell receptor for PEDV was the aminopeptidase N (APN) receptor. APN is a cell surface enzyme that is present in the small intestine of pigs and is involved in the processing of peptides. Previous research has shown that PEDV utilizes both porcine and human aminopeptidase N (APN) as receptors to bind to the host cell membrane, thereby facilitating viral entry into the cell [12]. However, there is ongoing debate and conflicting evidence regarding the role of APN as a receptor for PEDV entry. Several studies suggest APN is not required for PEDV entry [14,15,17]. We investigated the effect of human APN on PEDV entry by transducing ANPEP knockout Huh7 cells with rVSV-ΔG-EGFP-S. Our results showed that loss of APN significantly reduced the infectivity of rVSV-ΔG-EGFP-S, indicating APN plays a critical role in PEDV entry. ANPEP knockout decreased the titer of PEDV SD about 10 fold, suggesting either ANPEP is not a functional receptor of PEDV or PEDV utilizes multiple receptors for its entry. Several other receptors have been identified as being used by PEDV for entry into host cells. However, the complexity of PEDV receptor usage and cellular entry requires further investigation.

Cytoplasmic tail truncation of viral glycoproteins is a common strategy to enhance pseudovirus formation. This approach eliminates steric interference that may occur between heterologous viral glycoproteins and the vector matrix or capsid proteins. Truncation of the last 19 amino acids of the cytoplasmic tail of the SARS-COV Spike protein has been shown to significantly enhance pseudotyping of VSV as well as increase infectivity and titers of lentiviral vector-based pseudovirus [38,39,40]. The impact of C-terminal truncation of the PEDV spike protein on the efficiency of pseudotyped VSV production was investigated in this study. However, in contrast to these findings, truncating the PEDV S protein actually decreased the production of pseudotyped VSV in this study. The original purpose of this truncation was to eliminate potential ER retention sequences in the cytoplasmic tail. However, in the case of the PEDV spike protein, truncation of the cytoplasmic tail resulted in a decrease in the efficiency of VSV pseudotyping. Several factors may contribute to this decrease. Truncation of the cytoplasmic tail of PEDV could potentially impact cell surface levels of spike protein, ectodomain conformation, or functions such as receptor binding or fusogenicity [40,52,53]. Therefore, it is necessary to further investigate the mechanisms behind the decreased incorporation and/or titer caused by truncation of the cytoplasmic tail. These investigations should focus on elucidating the effects of cytoplasmic tail truncation on cell surface expression levels as well as the potential alterations in receptor binding or fusogenicity. By gaining a deeper understanding of these mechanisms, we can uncover the underlying factors contributing to the observed decrease in pseudotyping efficiency.

In addition, 25-hydroxycholesterol is an oxidized derivative of cholesterol and has been identified as a potential antiviral compound against several enveloped viruses, such as HIV [54], Ebola [55], and Zika [56]. Recent studies have suggested that 25HC can inhibit the entry and replication of coronaviruses, including PEDV, SARS-CoV-2, and MERS-CoV [42,43,44,45,57,58]. The mechanism by which 25HC exerts its antiviral effects on coronaviruses is not yet fully understood. However, it has been proposed that 25HC inhibits viral entry by interfering with the fusion process between the viral envelope and host cell membranes. This inhibition could occur through the disruption of lipid rafts, which are cholesterol-rich microdomains involved in viral entry. In this study, Huh7 cells were pretreated with different concentrations of 25HC and subsequently infected with the rVSV-ΔG-EGFP-S virus. The results showed a significant reduction in virus infection in 25HC-treated cells compared to untreated cells, as observed through fluorescent microscopy and flow cytometry analysis. These findings provide additional evidence for the potential usefulness of rVSV-ΔG-EGFP-S as a valuable tool for studying the inhibition of PEDV entry.

## 5. Conclusions

This study successfully generated a replication-deficient PEDV S pseudotyped VSV to study PEDV entry. The pseudotyped virus exhibited efficient infection capabilities in permissive cell lines, such as Vero, Huh7, and JHH7 cells, while resistance was observed in other cell lines. Truncation of the cytoplasmic tail of the PEDV spike protein was found to decrease the efficiency of viral pseudotyping, highlighting the importance of this region for proper pseudotyped VSV production. The inhibitory effects of 25HC on the entry of the pseudotyped virus further support its potential as an antiviral agent against PEDV. Additionally, the study demonstrated the involvement of human APN in facilitating PEDV entry. Furthermore, the ability of the PEDV S pseudotyped virus to infect swine intestinal organoids provides a valuable model for studying PEDV entry in a physiologically relevant environment. Collectively, the replication-deficient PEDV S pseudotyped virus serves as a versatile and valuable tool for investigating the complex dynamics of PEDV entry and infection, ultimately aiding in the development of effective control strategies for this economically significant virus.

## Figures and Tables

**Figure 1 microorganisms-11-02075-f001:**
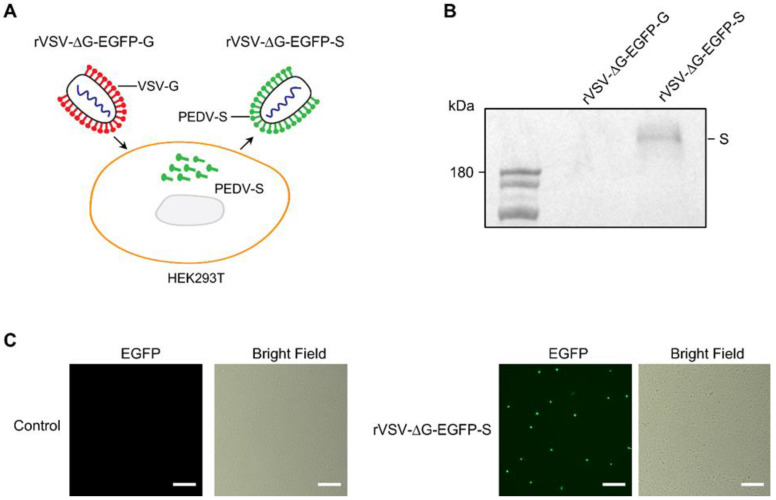
Generation and characterization of replication-deficient PEDV S pseudotyped rVSV-ΔG-EGFP-S virus. (**A**) Schematic diagram illustrates the process for producing pseudotyped viruses of PEDV. HEK-293T cells were transfected with PEDV spike expression vector pCAGGS-PEDV-S. At 24 h after transfection, cells were infected with VSV G pseudotyped virus (rVSV-∆G-eGFP-G). Upon viral infection, the uncoated rVSV-∆G-eGFP-G genome expresses all the enzymes and structural proteins from the VSV genome, except for the G protein, and undergoes partial genome replication. The expressed structural proteins and partially replicated genomes self-assemble into virus particles that lack an envelope. The virus is subsequently released from the host cells through budding, where the PEDV S protein expressed on the host cell surface becomes the envelope protein of the pseudovirion. As a result, the pseudotyped virus incorporates the PEDV S protein on the envelope, packaging a defective VSV genome containing a reporter GFP gene in its capsid. (**B**) The surface proteins of the virus particles were analyzed using western blotting. From left to right lane, showing control supernatant and PEDV pseudovirus supernatant. The control supernatant was prepared using the same procedure and used as a negative control. (**C**) Representative images of control (left panel) or rVSV-∆G-EGFP-S-infected (right panel) Vero cells. The GFP signal is shown in green. Scale bar: 200 μm.

**Figure 2 microorganisms-11-02075-f002:**
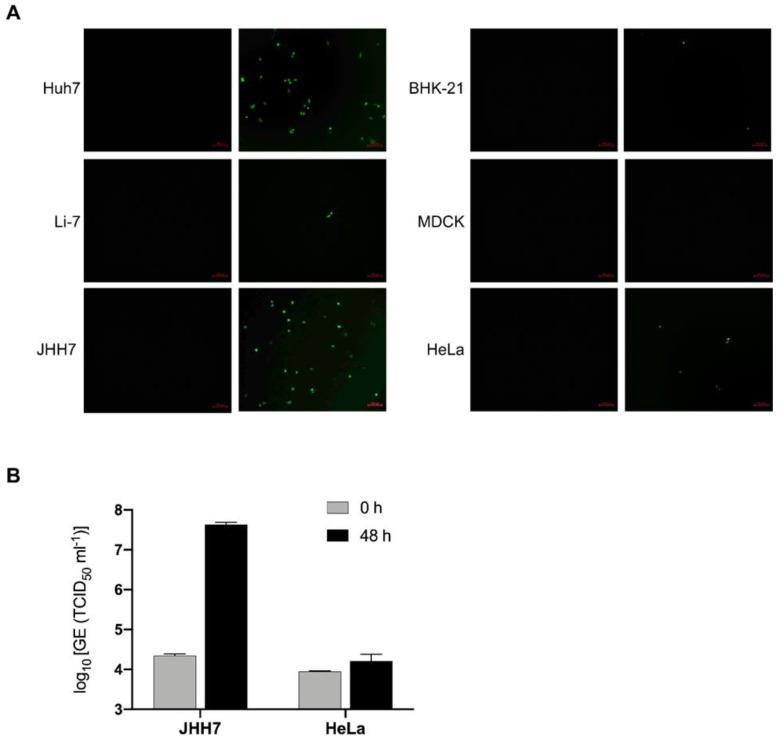
Characterization of rVSV-ΔG-EGFP-S entry on different cell lines. (**A**) Indicated cell lines were transduced with equal amounts (0.05 MOI) of rVSV-ΔG-EGFP-S virus. The images were captured by fluorescent microscopy at 12 hpi. Representative images are displayed. Scale bar: 100 μm. (**B**) The PEDV RNA levels in supernatants of JHH7 and HeLa cells infected with PEDV SD were quantified by RT-qPCR at 0 and 48 h. The results were expressed as genome equivalents (GE; TCID_50_ per mL). Error bars indicate standard deviation (SD).

**Figure 3 microorganisms-11-02075-f003:**
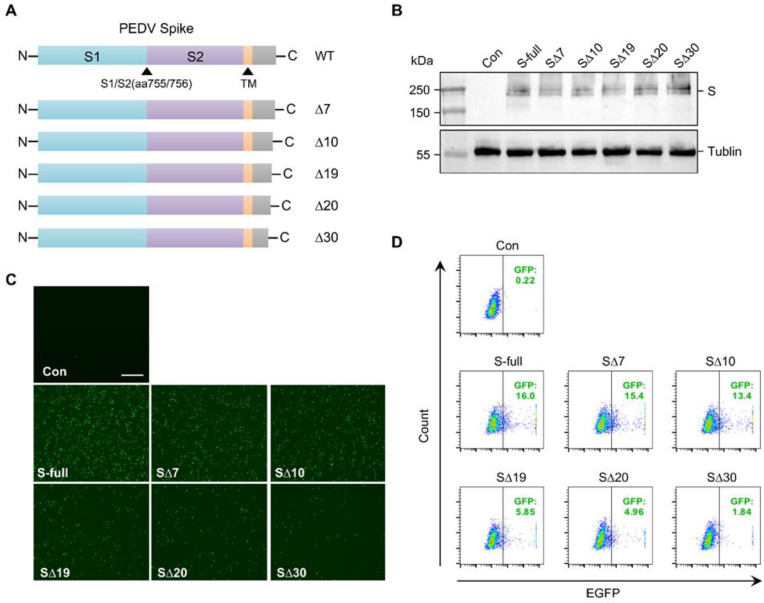
C-terminal truncation of PEDV spike decreases the efficiency of viral pseudotyping. (**A**) Schematic presentation of spike proteins of PEDV and the various C-terminal truncations of spike. The S1 and S2 subunits (The border between the S1 and S2 subunits was deduced from the sequence alignment with alphacoronavirus S proteins) and the transmembrane domain (TM predicted by TMHMM server) are indicated. (**B**) HEK-293T cells were transfected with same amount of various C-terminal truncations as well as full-length PEDV S vector. The expressions of C-terminal truncated S proteins were assessed by Western blotting. (**C**) The Vero cells were transduced with equal amount of supernatant collected from HEK-293T, which were transfected with different C-terminal truncations of S and infected with equal amount of VSV G pseudotyped virus (rVSV-∆G-eGFP-G). The images were captured by fluorescent microscopy at 12 hpi. The representative images are shown. Scale bar: 500 μm. (**D**) Samples from (**C**) were collected at 12 hpi to be analyzed by flow cytometry. The representative images were showing.

**Figure 4 microorganisms-11-02075-f004:**
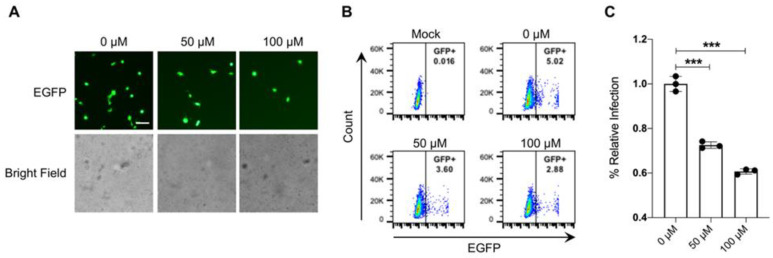
25-hydroxycholesterol could inhibit the pseudovirus rVSV-ΔG-EGFP-S entry. Huh7 cells were pretreated with different concentrations of 25HC for 2 h. And then Huh7 cells were infected with rVSV-ΔG-EGFP-S. (**A**) Microscopic images showing the Huh7 cells infected with rVSV-ΔG-EGFP-S (top panels) and Bright field images (bottom panels). Scale bars: 100 μm. (B) Samples were collected at 12 hpi to be processed by flow cytometry. (**C**) Corresponding analysis of flow cytometry histograms in (**B**). Data are expressed as %infection relative to DMSO-treated Huh7 cells. *** *p* < 0.001; *n* = 3.

**Figure 5 microorganisms-11-02075-f005:**
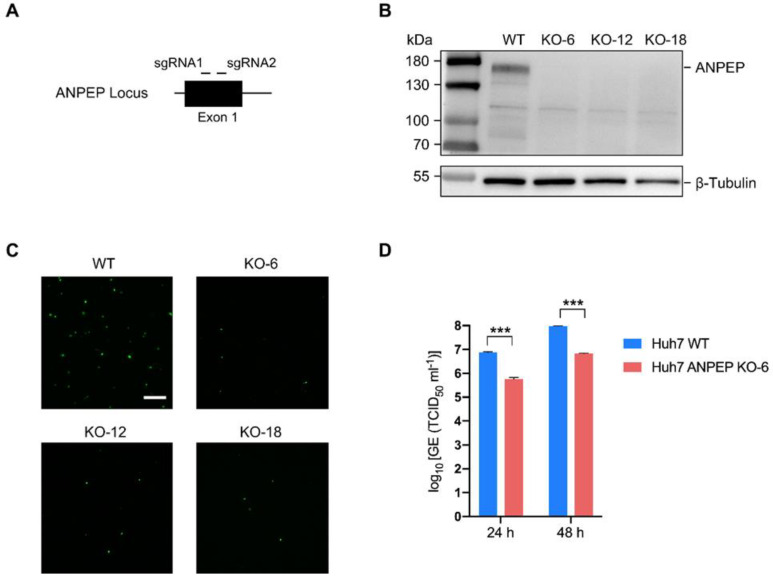
Human ANPEP could facilitate PEDV entry. (**A**) Schematic illustration of human ANPEP gene locus and the locations of two small guide RNAs. (**B**) ANPEP expression was determined by Western blotting in ANPEP-WT Huh7 cells and three ANPEP-KO Huh7 lines. (**C**) Representative images of ANPEP-WT and ANPEP-KO Huh7 were infected equal amount of rVSV-ΔG-EGFP-S at 12 hpi. Scale bar: 200 μm (**D**) The PEDV RNA levels in supernatants of the infected ANPEP-WT and one of ANPEP-KO cells at 24 and 48 h after infection with PEDV SD were quantified using a RT-qPCR assay and expressed as genome equivalents (GE; TCID_50_ per mL). Error bars indicate SD. *** *p* < 0.001; *n* = 5.

**Figure 6 microorganisms-11-02075-f006:**
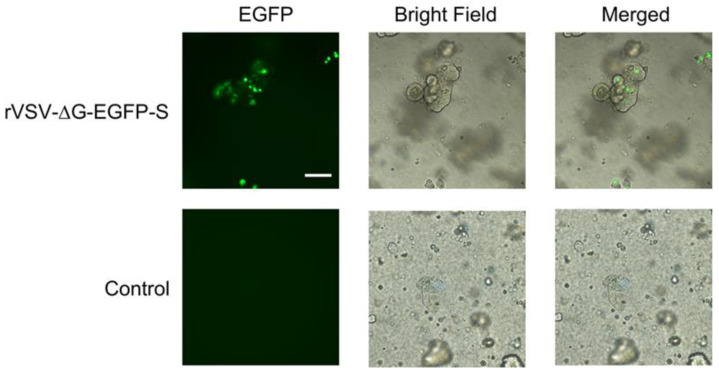
rVSV-∆G-EGFP-S pseudoviruses infect the porcine intestinal organoids. Swine intestinal organoids were transduced with equal amounts of rVSV-∆G-EGFP-S or control vectors with no spike protein. GFP expression was visualized 24 h after transduction. Scale bars: 100 μm.

**Table 1 microorganisms-11-02075-t001:** Oligo nucleotide sequences used in this study.

Oligo Name	Sequence (5′-3′)	Purpose
PEDV-S-F	CCGGAATTCATGACCCCCCTGATC	PCR
PEDV-S-R	CCGCTCGAGTCACTGCACGTGCAC	PCR
PEDV-SCΔ7-R	CCGCTCGAGTCAGGCTTCGTA	Truncation PCR
PEDV-SCΔ10-R	CCGCTCGAGTCAGGGCTGCAGTCTA	Truncation PCR
PEDV-SCΔ19-R	CCGCTCGAGTCAGCCACTAAAGCA	Truncation PCR
PEDV-SCΔ20-R	CCGCTCGAGTCAACTAAAGCAGGCG	Truncation PCR
PEDV-SCΔ30-R	CCGCTCGAGTCACCCGCAACAG	Truncation PCR
ANPEP-sgRNA-1	CCTTGGACCAAAGTAAAGCG	Knockout
ANPEP-sgRNA-2	ACGGGGTGGTGGAGGCCACG	Knockout
ANPEP-ID-F	CTGCAGCCTGTAACCAGACA	Knockout PCR
ANPEP-ID-R	GTCATTGGGGGTGAGGTACG	Knockout PCR
PEDV-qPCR-F	GAAGGCGCAAAGACTGAACC	Virus titer
PEDV-qPCR-R	TTGCCATTGCCACGACTCCT	Virus titer

## Data Availability

The data that support the findings of this study are available from the corresponding author upon reasonable request.

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
