# Peer review of "Establishment of Replication Deficient Vesicular Stomatitis Virus for Studies of PEDV Spike-Mediated Cell Entry and Its Inhibition"

_microorganisms, 2023, doi:10.3390/microorganisms11082075_

Round 1
Reviewer 1 Report
Luo and Cols used an S protein of PEDV to produce recombinant pseudotyped vesicular stomatitis virus (VSV) particles (rVSV-@G-EGEP-S). The S protein pseudotype VSV virus was correctly produced and infects several cell lines. This pseudotype VSV virus was successfully produced without removing the cytoplasmatic region, contrary to other Coronavirus. This pseudotype VSV virus was also used in infection experiments using cell lines and intestinal organoids.
The study is interesting and with the merits to be published in this Journal.
I have some comments for the authors expecting to increase the clarity of the study.
A low infection rate of the PEDV S pseudotyped virus is observed in Figure 2. However, the results of Figure 3 using the Vero cell are significantly different. Including Vero cells in Figure 2 will be very illustrative for the reader.
Section 3.5. there is a problem with the description and figures of this section, and it is hard to follow. Please correct accordingly.
Section 3.6. A marker to confirm the intestinal organoids will increase the quality of the result.
Line 381. The statement is wrong; please correct accordingly. It is high compared with other cell lines but not higher than Vero cells.
None
Author Response
We appreciate the time and effort that the reviewer has put into providing critiques of our manuscript. We have carefully considered all of the comments and have made the necessary changes to the text to address the reviewer's concerns. The modifications have been made throughout the manuscript to ensure that all relevant sections have been updated accordingly. The followings are our point-by-point responses:
Reviewer comments:
Luo and Cols used an S protein of PEDV to produce recombinant pseudotyped vesicular stomatitis virus (VSV) particles (rVSV-@G-EGEP-S). The S protein pseudotype VSV virus was correctly produced and infects several cell lines. This pseudotype VSV virus was successfully produced without removing the cytoplasmatic region, contrary to other Coronavirus. This pseudotype VSV virus was also used in infection experiments using cell lines and intestinal organoids.
The study is interesting and with the merits to be published in this Journal.
I have some comments for the authors expecting to increase the clarity of the study.
A low infection rate of the PEDV S pseudotyped virus is observed in Figure 2. However, the results of Figure 3 using the Vero cell are significantly different. Including Vero cells in Figure 2 will be very illustrative for the reader.
Response: Thanks for your suggestion. Vero cells that were infected with PEDV S pseudotyped VSV at an MOI of 0.01 have been included in Figure 1C. The amount of pseudovirus used in this experiment was the same as in Figure 2A. In Figure 3, our objective was to compare the impact of the C-terminal truncated spike on VSV pseudotyping efficiency. To achieve this, we utilized an equivalent amount of PEDV S pseudotyped virus. Specifically, we added one milliliter of virus-containing supernatant to each well of a six-well plate for infection. Consequently, this approach resulted in an MOI of approximately 0.5, effectively increasing the infection rate, as depicted in Figure 3.
Section 3.5. there is a problem with the description and figures of this section, and it is hard to follow. Please correct accordingly.
Response: Thanks for your careful review. We have made a switch between Figure 4 and Figure 5, and we have ensured that the figures are correctly cited.
Section 3.6. A marker to confirm the intestinal organoids will increase the quality of the result.
Response: In this study, we used intestinal organoids from the same batch that was used in a previous study. These organoids have already been characterized in the published paper, which provides an important context for our current work (1).
Line 381. The statement is wrong; please correct accordingly. It is high compared with other cell lines but not higher than Vero cells.
Response: As suggested by the reviewer, we have polished the statement as “Among the tested cell lines, Huh7 and JHH7 cells were susceptible to the PEDV spike pseudotyped virus, whereas BHK-21, HeLa, MDCK, and Li7 cells showed resistance. Interestingly, JHH7 cells exhibited susceptibility to the PEDV spike pseudotyped VSV, which was first reported. This finding is significant because it suggests that JHH7 cells may be useful for studying the entry mechanism of PEDV.”
Reference:
- Zhang M, Lv L, Cai H, Li Y, Gao F, Yu L, Jiang Y, Tong W, Li L, Li G, Tong G, Liu C. 2022. Long-Term Expansion of Porcine Intestinal Organoids Serves as an in vitro Model for Swine Enteric Coronavirus Infection. Front Microbiol 13:865336.
Reviewer 2 Report
This manuscript generated a PEDV spike pseudotyped VSV and investigated pseudovirus entry using 6 cell lines and intestinal organoids. As the exact cell receptors and/or contributions of numerous candidate receptors for PEDV entry have not been demonstrated explicitly, such a pseudovirus system should be useful for this line of research. I suggest the following points for authors to consider:
1. Brief description of each of the cell lines used should be added.
2. The demonstration of reduced pseudotyping upon truncation of the spike protein is interesting. However, whether this reduction was due to different levels of the spike proteins (Figure 3B) rather than incorporation efficiency into the VSV pseudovirus requires clarification. I would suggest the authors to normalize pseudovirus titres against the intracellular spike protein levels. If the conclusion still holds true, potential mechanisms should be discussed in the discussion. For instance, in comparison to those of SARS-CoVs, are there any amino acid residues or motifs in the PEDV spike cytoplasmic tail that may be particularly important for this function?
3. Figures 4 and 5 should be switched.
Author Response
We appreciate the thorough review and constructive suggestions. Please find below our point-by-point responses:
Reviewer comments:
This manuscript generated a PEDV spike pseudotyped VSV and investigated pseudovirus entry using 6 cell lines and intestinal organoids. As the exact cell receptors and/or contributions of numerous candidate receptors for PEDV entry have not been demonstrated explicitly, such a pseudovirus system should be useful for this line of research. I suggest the following points for authors to consider:
1. Brief description of each of the cell lines used should be added.
Response: In the "Materials and Methods" section, we have included a brief description of each cell line: The Huh7, JHH7, and Li7 cell lines, derived from human hepatocellular carcinoma, exhibit an epithelial-like morphology. The BHK-21 cell line, on the other hand, originates from a baby hamster's kidney. Additionally, the MDCK (Madin-Darby Canine Kidney) cell line is widely used in cell biology and virology research and is derived from the kidney of a female cocker spaniel.
2. The demonstration of reduced pseudotyping upon truncation of the spike protein is interesting. However, whether this reduction was due to different levels of the spike proteins (Figure 3B) rather than incorporation efficiency into the VSV pseudovirus requires clarification. I would suggest the authors to normalize pseudovirus titres against the intracellular spike protein levels. If the conclusion still holds true, potential mechanisms should be discussed in the discussion. For instance, in comparison to those of SARS-CoVs, are there any amino acid residues or motifs in the PEDV spike cytoplasmic tail that may be particularly important for this function?
Response: Thank you for your critical comment. We greatly appreciate your suggestion to normalize pseudovirus titers against the intracellular spike protein levels. This normalization will help elucidate whether the decrease in pseudotyping is attributable to varying spike protein levels or differences in incorporation efficiency into the VSV pseudovirus. We are confident in the validity of our conclusion for the following reasons. Firstly, we repeated all transfections with C-terminal truncated PEDV at least three times, consistently yielding similar results. for this article, we have presented only a representative outcome. Additionally, as suggested by the reviewer, we conducted a quantitative analysis of the Western blot results to normalize the levels of intracellular spike protein. The expressed levels of intracellular spike protein in the 293T cells were found to be comparable (Figure 3B).
In response to the reviewer's suggestion, we have included a discussion of potential mechanisms in the “Discussion” section: Cytoplasmic tail truncation of viral glycoproteins is a common strategy to enhance pseudovirus formation. This approach eliminates steric interference that may occur between heterologous viral glycoproteins and the vector matrix or capsid proteins. Truncation of the last 19 amino acids of the cytoplasmic tail of the SARS-COV Spike protein has been shown to significantly enhance pseudotyping of VSV, as well as increase infectivity and titers of lentiviral vector-based pseudovirus (1-3). The original purpose of this truncation was to eliminate potential ER retention sequences in the cytoplasmic tail. However, in the case of the PEDV spike protein, truncation of the cytoplasmic tail resulted in a decrease in the efficiency of VSV pseudotyping. Several factors may contribute to this decrease. Truncation of the cytoplasmic tail of PEDV could potentially impact cell surface levels of spike protein, ectodomain conformation, or functions such as receptor binding or fusogenicity(2, 4, 5). Therefore, it is necessary to further investigate the mechanisms behind the decreased incorporation and/or titer caused by truncation of the cytoplasmic tail. These investigations should focus on elucidating the effects of cytoplasmic tail truncation on cell surface expression levels, as well as the potential alterations in receptor binding or fusogenicity. By gaining a deeper understanding of these mechanisms, we can uncover the underlying factors contributing to the observed decrease in pseudotyping efficiency.
3. Figures 4 and 5 should be switched.
Response: Thanks for your careful review. We have made a switch between Figure 4 and Figure 5, and we have ensured that the figures are correctly cited.
Reference:
- Chen HY, Huang C, Tian L, Huang X, Zhang C, Llewellyn GN, Rogers GL, Andresen K, O'Gorman MRG, Chen YW, Cannon PM. 2021. Cytoplasmic Tail Truncation of SARS-CoV-2 Spike Protein Enhances Titer of Pseudotyped Vectors but Masks the Effect of the D614G Mutation. J Virol 95:e0096621.
- Yu J, Li Z, He X, Gebre MS, Bondzie EA, Wan H, Jacob-Dolan C, Martinez DR, Nkolola JP, Baric RS, Barouch DH. 2021. Deletion of the SARS-CoV-2 Spike Cytoplasmic Tail Increases Infectivity in Pseudovirus Neutralization Assays. J Virol 95.
- Johnson MC, Lyddon TD, Suarez R, Salcedo B, LePique M, Graham M, Ricana C, Robinson C, Ritter DG. 2020. Optimized Pseudotyping Conditions for the SARS-COV-2 Spike Glycoprotein. J Virol 94.
- Chen J, Kovacs JM, Peng H, Rits-Volloch S, Lu J, Park D, Zablowsky E, Seaman MS, Chen B. 2015. HIV-1 ENVELOPE. Effect of the cytoplasmic domain on antigenic characteristics of HIV-1 envelope glycoprotein. Science 349:191-5.
- Cathomen T, Naim HY, Cattaneo R. 1998. Measles viruses with altered envelope protein cytoplasmic tails gain cell fusion competence. J Virol 72:1224-34.